# Employee Proactive Personality and Career Growth: The Role of Proactive Behavior and Leader Proactive Personality

**DOI:** 10.3390/bs14030256

**Published:** 2024-03-21

**Authors:** Guimei Ma, Xianru Zhu, Bing Ma, Hermann Lassleben

**Affiliations:** 1School of Management, Xi’an Polytechnic University, Xi’an 710048, China; 20151205@xpu.edu.cn (G.M.); 210511058@stu.xpu.edu.cn (X.Z.); 2ESB Business School, Reutlingen University, 72762 Reutlingen, Germany; hermann.lassleben@reutlingen-university.de

**Keywords:** proactive personality, voice behavior, taking charge, career growth

## Abstract

Based on social information processing theory, this research examines whether and how an employee’s proactive personality influences intrinsic and extrinsic career growth. It also examines the mediating effects of two types of proactive behaviors (voice behavior and taking charge) and the moderating effect of a leader’s proactive personality. A sample of 307 employee-leader dyads participated in this survey. Structural equation modeling was used to test the hypotheses, and the bootstrap procedure was used to test the indirect effects. Results show that an employee’s proactive personality has significant positive effects on both intrinsic and extrinsic career growth. The mediating effect of taking charge was confirmed, while the mediating effect of voice behavior was not. Leader proactive personality weakens the relationship between employee proactive personality and the two types of proactive behaviors. Employee proactive personality is more positively related to intrinsic and extrinsic career growth via proactive behaviors when a leader’s proactive personality is low. This study extends the literature on proactive personality, proactive behavior, and career development by examining the underlying determination, mediation, and moderation mechanisms.

## 1. Introduction

In the current era of boundaryless career development, organizational career growth, which refers to an individual’s career development and advancement within an organization [1,2], has attracted the attention of organizations as a way to prevent talent loss and improve organizational competitiveness [3]. Organizational career growth can be divided into intrinsic (career goal progress and professional ability development) and extrinsic (promotion speed and remuneration growth) career growth [4]. Previous studies have explored the impact of personality traits on career growth [5], but only focused on the Big Five personality [6]. In fact, the proactive personality trait outperforms the Big Five personality traits in predicting numerous workplace phenomena [7]. However, less attention has been paid to employees’ proactive personality as an antecedent of intrinsic and extrinsic organizational career growth, especially combined with the boundary role of leaders’ proactive personality.

Social information processing theory posits that individuals rely on social cues and information to form a picture of their environment that guides their behavior [8]. As receivers of social information, proactive employees will collect and process contextual cues from organizations and leaders to shape their cognitive processes, ultimately enabling intrinsic and extrinsic career growth. Meanwhile, in the process of social information processing, employees make behavioral choices, such as engaging in voice behavior and/or taking charge, based on processing information from the social/organizational environment. Thus, we expect that proactive employees who process information about organizational shortcomings will propose (voice) and/or implement (take charge) changes to improve organizational functionality, which will also affect their career outcomes [9,10], i.e., support their intrinsic and extrinsic career growth.

Based on the above, the aim of this study is to examine the effect of employee proactive personality on intrinsic and extrinsic career growth and to explore the mediating role of voice behavior and taking charge as well as the moderating role of leader proactive personality. In doing so, this study makes three important theoretical contributions. First, it contributes to the organizational career growth literature by examining the effects of employee proactive personality on intrinsic and extrinsic career growth. Thereby, it broadens the understanding of the determinants of organizational career growth and extends the multidimensional view of organizational career growth. Second, it extends our knowledge of the consequences of employee proactive behavior by examining the mediating effects of voice behavior and taking charge of the relationship between employee proactive personality and organizational career growth. Third, it goes beyond employee-centered models of proactivity by exploring the role of a leader’s proactive personality as a key contextual variable for the hypothesized relationship. In doing so, we advance the understanding of the moderating role of a leader’s proactive personality from the perspective of social information processing and extend the research of proactive personality. The complete conceptual model is shown in Figure 1.

## 2. Theory and Hypotheses

### 2.1. Social Information Processing Theory

Social information processing theory posits that individuals rely on social cues and information to form a picture of their environment that guides their behavior [8]. Zalesny and Ford (1991) extended the social information processing perspective to include personal and situational factors that influence how individuals process social information [11]. According to social information processing theory, social context and social relationships play an important role in shaping individuals’ cognitive processes and influencing their behavior. Applied to organizations, this means that employees seek and use information from their social networks, i.e., they rely on the perceptions, attitudes, and behaviors of social contacts, especially managers and coworkers, as reference points for interpreting ambiguous situations, reducing uncertainty, and making decisions. Although seemingly simple, the social information processing model highlights how complex social and cognitive processes interact and influence the formation of attitudes and beliefs that underlie the behavioral choices people make in response to social/organizational situations [8]. In this research, as receivers of social cues and information, proactive employees will process these contextual cues to express their attitudes and behavioral tendency to create constructive environmental changes (i.e., voice behavior and taking charge). Managers and peers can also assess these behaviors and respond accordingly by processing social information related to employees’ behavioral tendency. Finally, these proactive employees can realize intrinsic and extrinsic career growth after information assessment by themselves and their leaders.

### 2.2. The Linkage between Employee Proactive Personality and Career Growth

Proactive personality is considered a relatively stable personality trait and describes the tendency of individuals to interact with their environment in ways that lead to constructive change [12,13]. Proactive employees have the potential and willingness to deal effectively with occupational constraints. They are sensitive and attentive to social cues, especially those related to organizational problems and development opportunities [14]. As a result, they can more easily locate, collect, encode, and analyze information needed to effect organizational change and support personal development [15], ultimately enabling intrinsic and extrinsic career growth.

We expect that employees’ proactive personality contributes to their intrinsic career growth. Employees with high proactive personalities not only show more initiative but also see and process more information that signals problems in the workplace than employees with low proactive personalities [9]. The drive to address and solve these problems, as well as the motivation to continuously improve their job skills according to the organization’s requirements, guides their behavior [16,17]. They invest in acquiring new knowledge and learning new professional skills and transform them into competencies that are valuable to the organization [13]. This benefits both, the success of the organization and the achievement of individual career goals. To explain upward mobility, the literature distinguishes between the contest mobility model and the sponsored mobility model [18]. According to the contest mobility model, individuals compete for upward mobility opportunities and realize career development based on their abilities, efforts, and attributes [19]. Therefore, we expect that proactive employees, driven by organizational information, continuously refine their skills and increase their effectiveness, thereby realizing intrinsic career growth.

We also expect that employees’ proactive personality contributes to their extrinsic career growth. When employees, driven by their proactive personality, engage in improving work practices, they demonstrate personality traits such as commitment, initiative, perseverance, and self-discipline. They demonstrate their competence and willingness to work voluntarily and constructively to improve their work environment [20]. Perceptions or information about these behaviors contribute to the appreciation and trust of managers and peers [17]. In return, employees receive recognition and praise, accumulate social capital, and receive sponsorship [21]. According to the sponsored mobility model, sponsored employees receive more development opportunities, including promotions and salary increases [22], which contribute to their extrinsic career growth [8]. In summary, our hypotheses are as follows:

**Hypothesis 1a.** 
*Employees proactive personality is positively associated with intrinsic career growth.*


**Hypothesis 1b.** 
*Employees proactive personality is positively associated with extrinsic career growth.*


### 2.3. The Mediating Role of Voice Behavior

Voice behavior refers to the “discretionary communication of ideas, suggestions, concerns, or opinions about work-related issues with the intent to improve organizational or unit functioning” [23] (p. 375). According to social information processing theory, the organizational environment is an important source of information that influences employees’ perceptions, attitudes, and behaviors. Proactive employees understand how to read the organizational environment, can identify opportunities and threats, and are able to categorize and respond to them, including using voice behavior when appropriate [24]. Previous empirical studies have found that employee proactive personality is positively related to voice behavior [9,16].

Because they make constructive suggestions to improve organizational processes and work practices [16], we expect the use of voice behavior to contribute to employees’ intrinsic career growth. By expressing ideas to improve the work environment, employees signal a positive attitude toward work [25,26]. Managers and peers who receive such cues are open to exchanging ideas and thoughts, which allows ‘voicers’ to acquire additional professional and organizational knowledge, thereby enhancing their professional capabilities and ultimately helping them to become clearer about their own career paths and goals [19,27].

In addition, we expect that voice behavior will also contribute to employees’ extrinsic career growth. By exhibiting voice behavior, employees signal competence, identification, and passion, which not only strengthens their relationships with managers and colleagues [28,29] but also obtains their support and career guidance [25], i.e., social capital that facilitates promotions and salary increases according to the sponsored mobility model. Previous empirical research has confirmed that managers evaluate employees’ promotability by processing social information, including voice behavior [9,30]. Therefore, we propose the following hypotheses:

**Hypothesis 2a.** 
*Voice behavior mediates the relationship between employees’ proactive personality and intrinsic career growth.*


**Hypothesis 2b.** 
*Voice behavior mediates the relationship between employees’ proactive personality and extrinsic career growth.*


### 2.4. The Mediating Role of Taking Charge

In contrast to voice behavior, which refers to the communication of constructive ideas to improve the state of the organization, taking charge refers to concrete actions to bring about functional change in the organization [31,32]. According to social information processing theory, the organizational environment attracts employees’ attention and provides cues that they use to interpret events, form attitudes, and determine their actions [10]. Proactive employees tend to act as change agents who challenge the status quo and address and solve problems they identify without being asked to ensure the functioning of the organization as well as their own development. They are not only committed to the organization’s goals but actively seek opportunities to engage in taking charge [20]. Zhang et al. (2023) have supported the positive relationship between employee proactive personality and taking charge in empirical research [17].

In addition to bringing about functional changes in the work environment, taking charge also affects employees’ organizational career growth. Taking charge is expected to contribute to intrinsic career growth by enhancing the professional skills needed to achieve career goals. Actively promoting new technologies, structures, and policies contributes to the development of employees’ skills [33]. It also builds on and fosters passion and motivation. Employees who take charge bring physical, cognitive, and emotional energy to their work. Consistent with social information processing theory, managers and colleagues who perceive these cues will value the behavior and respond accordingly [9]. Simply put, proactive employees make voluntary and constructive efforts to bring about change [34]. In doing so, they not only develop further valuable professional skills, but also receive affirmation, recognition, and praise from managers and peers, which in turn opens up additional professional development opportunities that lead to intrinsic career growth.

Taking charge can also be expected to contribute to extrinsic career growth by earning recognition and helping employees build new relationships. Employees who take charge must plan, coordinate, and communicate the changes they seek. As a result, they take a leadership role in change, which earns them the respect of other organizational members [10,35]. When they need advice or support from managers or colleagues during the process, collaborative relationships are established or strengthened. Recognition and the accumulation of social capital, such as status and interpersonal relationships, in turn, affect promotion and compensation according to the sponsored mobility model, thus contributing to extrinsic career growth [36]. In summary, our hypotheses are as follows:

**Hypothesis 3a.** 
*Taking charge mediates the relationship between employees’ proactive personality and intrinsic career growth.*


**Hypothesis 3b.** 
*Taking charge mediates the relationship between employees’ proactive personality and extrinsic career growth.*


### 2.5. The Moderating Role of Leaders’ Proactive Personality

According to social information processing theory, leaders’ behavioral orientations and preferences (e.g., personality dispositions) are important sources of information and situational conditions that influence employees’ perceptions of the work environment as well as their behaviors [8,10,37]. Therefore, we assume that leaders’ proactive personality can significantly influence the relationship between employees’ proactive personality and their proactive behaviors. Leaders with highly proactive personalities demonstrate competence and value orientation and expect their employees to also show initiative and responsibility for their environment [13]. In an effort to understand their leaders’ expectations, employees who correctly decode these cues will strive to behave accordingly (i.e., speak up and take charge) regardless of their personality [8,11,38]. Previous research has found an environment that encourages proactivity weakens the positive effect of a proactive personality on proactive behaviors [16]. More specifically, we expect that leaders’ proactive personalities may substitute the role of employees’ proactive personalities in influencing voice behavior and taking charge.

Employees with low proactive personalities lack the impulse to improve their work environment. Instead, they are more likely to react to environmental influences [38,39]. Under the influence of leaders’ proactive personality, they are more motivated to behave proactively, contrary to their personality structure. In detail, leaders with highly proactive personalities signal to employees that they expect proactive behavior by providing appropriate information and cues [40,41]. By asking them to challenge the status quo and showing them action plans to solve problems [7,40], they encourage them to engage in proactive behavior and to speak up and take charge. This is effective because employees who themselves are less proactive and tend to ‘wait and see’ are more receptive to social cues in the form of observed behavioral tendencies from their leaders [37]. Therefore, under a leader with a highly proactive personality, non-proactive employees are also more likely to show voice behavior and take charge. Employees can expect that speaking up and taking charge will be welcomed and rewarded and will lead to career benefits [37].

Overall, we expect informational cues from leaders’ proactive personalities to be more critical for less proactive employees, because proactive employees already have tendencies to behave proactively [16]. When leaders’ proactive personality is high, employees are likely to engage in proactive behaviors whether they are proactive or not. By contrast, when leaders’ proactive personality is low, employees’ motivation to engage in voice behavior and take charge depends more on their own inherent personality. Consequently, leaders’ proactive personality can at least partially replace the role of employees’ proactive personality in motivating them to speak up and take charge. Based on the above, we hypothesize the following:

**Hypothesis 4a.** 
*Leaders’ proactive personality moderates the positive relationship between employees’ proactive personality and voice behavior, such that the relationship is stronger when leaders’ proactive personality is low compared to high.*


**Hypothesis 4b.** 
*Leaders’ proactive personality moderates the positive relationship between employees’ proactive personality and taking charge, such that the relationship is stronger when leaders’ proactive personality is low compared to a high.*


Following and integrating the above argumentation, we further expect a moderated mediation effect in which leaders’ proactive personality also plays a moderating role in the indirect relationship between employees’ proactive personality and their intrinsic and extrinsic career growth through proactive behaviors, i.e., voice behavior and taking charge. As discussed above, we expect that employees with low proactive personalities will increase their proactive behaviors when they interpret their leaders’ proactive personality as a cue for corresponding expectations about their own behavior, thereby improving their standing with the leader [37,42]. In turn, the voice behavior and taking charge that employees demonstrate influence how leaders evaluate those employees [9,17]. The more trust, respect, and support employees receive from leaders, the more likely they are to realize intrinsic career growth through the acquisition of professional skills and the achievement of career goals, as well as extrinsic career growth through the accumulation of social capital that facilitates promotions and compensation increases. In this regard, the proactive personality of leaders not only substitutes the role of employee proactive personality in terms of promoting less proactive employees to engage in voice behavior and taking charge but also in terms of facilitating their organizational career growth. Therefore, our final hypotheses are as follows:

**Hypothesis 5a.** 
*Leaders’ proactive personality moderates the indirect relationship between employees’ proactive personality and intrinsic and extrinsic career growth via voice behavior, such that the relationship is stronger when leaders’ proactive personality is low compared to high.*


**Hypothesis 5b.** 
*Leaders’ proactive personality moderates the indirect relationship between employees’ proactive personality and intrinsic and extrinsic career growth via taking charge, such that the relationship is stronger when leaders’ proactive personality is low compared to high.*


## 3. Methods

### 3.1. Participants and Procedures

A sample was drawn from different companies mainly in five regions of China (Shaanxi, Shanghai, Zhejiang, Guangdong, and Beijing), including state-owned enterprises, private enterprises, wholly foreign-owned enterprises, and so on. To ensure the universality of the sample, we chose companies from various industries. Chinese-language questionnaires were pre-coded to allow the matching of subordinates with leaders. The subordinate questionnaires included a self-report measure of the respondent’s proactive personality and organizational career growth, while the leader questionnaires included a self-report measure of the leader’s proactive personality as well as measures of voice behavior and taking charge of the paired employees. The study was approved by the Research Ethics Committee of the School of Management of Xi’an Polytechnic University.

Before the start of the study, we approached organizational managers in our social networks through telephone, and specifically communicated with them about the study goals, the needed support and cooperation, as well as the distribution method and arrangement of questionnaires. And then, we arrived at the companies at the agreed time. With the help of managers, informed consent forms and questionnaires were distributed to subordinates and their supervisors working in research and development, marketing, production, human resources, finance, and other departments. Each supervisor was paired with one or two subordinates. Participants were assured that their responses would be kept confidential and used only for research purposes. To encourage participation, participants received small gifts as compensation.

Of the 430 sets of questionnaires distributed, 335 were collected. After omitting blank, invalidly paired, and obviously incorrect questionnaires, the final sample consisted of 307 sets of supervisor/subordinate responses.

### 3.2. Measures

All variables included in the conceptual model (Figure 1) were assessed using validated scales from the literature and were widely used in the past. Items were measured on a 7-point Likert scale (1 = strongly disagree, 7 = strongly agree). Since all scales were originally in English, items were translated into Chinese using the back-translation method to ensure a high degree of accuracy.

The proactive personality of employees and leaders was measured with six items developed by Parker (1998) [43]. A sample item reads: “If I believe in an idea, no obstacle will prevent me from making it happen”. Previous studies have shown that the scale was reliable and had good validity with Chinese samples [17]. In this study, Cronbach’s alpha was 0.88 for employees and 0.83 for leaders.

Voice behavior was assessed using Dyne and Lepine’s (1998) 6-item scale [44], which is often used in the Chinese context [45]. A sample item reads: “This staff speaks up in the group with ideas for new projects or changes in procedures”. Cronbach’s alpha for this scale was 0.90.

Taking charge was measured using Morrison and Phelps’s (1999) 9-item scale [46]. Sample items read: “This staff often tries to change organizational rules or policies that are nonproductive or counterproductive”, and “This staff often tries to correct a faulty procedure or practice”. Cronbach’s alpha for this scale was 0.95.

Organizational career growth was measured using a scale developed by Weng, McElroy, Morrow, and Liu (2010) [47]. It consists of 15 items representing four domains: professional ability development (4 items), career goal achievement (4 items), promotion speed (4 items), and remuneration growth (3 items). The eight items in the first two domains together measure intrinsic career growth, and the seven items in the last two domains measure extrinsic career growth. The scale showed good reliability and validity in previous studies with Chinese samples [6]. Cronbach’s alpha was 0.93 for intrinsic career growth and 0.89 for extrinsic career growth.

Consistent with previous research [17,19,21], we controlled for demographic characteristics of employees, such as age, gender, education, and tenure, as these are individual-level influences on proactive behavior [25,31]. We also controlled for leaders’ gender, age, and the region where the company is located. Since previous studies have found that state ownership of firms leads to different organizational behavior [48], we also included company nature as a control variable.

### 3.3. Analytical Strategy

Confirmatory factor analyses (CFA) were conducted in Amos 24 to evaluate the conceptual model. Descriptive statistics (means, standard deviations, and correlations) were computed for all variables. Structural equation modeling was performed in Amos 24 to test the hypotheses (direct, indirect, and moderating effects). In addition, to test the hypothesized conditional indirect effects, based on previous studies [28,31], we used the PROCESS macro in SPSS (Model 7) to report the 95% confidence interval (CI).

## 4. Results

### 4.1. Confirmatory Factor Analyses

Confirmatory factor analyses were conducted in Amos 24 to test the discriminant validity of the variables. The results indicated that a six-factor model with employee proactive personality, leader proactive personality, voice behavior, taking charge, intrinsic, and extrinsic career growth was acceptable (χ^2^ = 1582.28, df = 800, CFI = 0.92, IFI = 0.93, TLI = 0.92, RMSEA = 0.06, SRMR = 0.04), while the fit indices of the alternative models were not acceptable. Thus, the results support the discriminant validity of the measurement model. They are presented in Table 1. In addition, we also estimated the average variance extracted (AVE) and composite reliabilities (CR). The results of the AVE were 0.55, 0.50, 0.65, 0.67, 0.72, and 0.65 for employee proactive personality, leader proactive personality, voice behavior, taking charge, intrinsic, and extrinsic career growth, respectively. The results of the CR were 0.88, 0.83, 0.92, 0.98, 0.95, and 0.93.

In this study, we examined the data’s common method bias (CMB) by creating a common method bias factor. The common method bias factor was included in the structural equation model. Results showed that the variation of the fit index as follows: ΔRMSEA = 0.01, ΔCFI = 0.02, ΔIFI = 0.01, and ΔTLI = 0.01. The change in each fit index was less than 0.03, which indicates that in this study did not exist serious CMB.

### 4.2. Demographic Characteristics of the Participants

The final sample consisted of 307 sets of supervisor/subordinate responses, with a response rate of 71%. 49.5% of the subordinates were male and 50.5% were female. Most were young, with 68.1% between the ages of 25 and 35. 51.8% had a bachelor’s degree, while 16.0% had a master’s degree or higher. The remainder had lower levels of education. 53.0% had more than three years of work experience. Among supervisors, 61.6% were male and 47.2% were between the ages of 35 and 50. The education level of supervisors was highest with a bachelor’s degree, accounting for 56.7%.

### 4.3. Hypotheses Testing

Means, standard deviations, and correlations among all study variables are presented in Table 2. Path analysis results are presented in Figure 2. Table 3 presents results for indirect and conditional indirect effects with bootstrapped 95% confidence intervals.

Previous research has recommended determining the model fit of structural equation models before adding and analyzing interaction effects [49]. Therefore, we ran a base model without the moderating variable, and the model fit well (χ^2^ = 1830.29, df = 880, CFI = 0.90, IFI = 0.90, TLI = 0.90, RMSEA = 0.06). Hypothesis 1a,b proposed that employee proactive personality would be positively related to organizational career growth. The path coefficients in Figure 2 show that employees’ proactive personality was positively related to intrinsic (B = 0.49, *p* < 0.001) and extrinsic (B = 0.34, *p* < 0.001) career growth, supporting Hypothesis 1a,b. Furthermore, we found that voice behavior was not related either to intrinsic (B = 0.10, *p* = 0.36) or extrinsic career growth (B = 0.01, *p* = 0.88). As shown in Table 3, the indirect effects of employee proactive personality on intrinsic (B = 0.05, SE = 0.06, 95% CI [−0.06, 0.17]) and extrinsic (B = 0.01, SE = 0.05, 95% CI [−0.08, 0.12]) career growth via voice behavior were not significant because the confidence interval includes zero. Therefore, Hypothesis 2a,b were not supported.

According to the results shown in Figure 2, taking charge is positively related to intrinsic (B = 0.30, *p* < 0.05) and extrinsic career growth (B = 0.25, *p* < 0.05). The results in Table 3 indicate that taking charge partially mediates the effects of employee proactive personality on intrinsic (B = 0.11, SE = 0.06, 95% CI [0.01, 0.23]) and extrinsic career growth (B = 0.09, SE = 0.05, 95% CI [0.01, 0.19]), supporting Hypothesis 3a,b.

Hypotheses 4 and 5 were tested using the moderated mediation model. Hypothesis 4a predicted that a leader’s proactive personality would moderate the relationship between employee proactive personality and voice behavior. As shown in Figure 2, the interaction between employee and leader proactive personality significantly predicted voice behavior (B = −0.14, SE = 0.06, *p* < 0.05), supporting Hypothesis 4a. The interaction is shown graphically in Figure 3. The slope difference tests showed that the positive relationship between employee proactive personality and voice behavior was stronger when leader proactive personality was low (simple slope = 0.35, *p* < 0.001) than high (simple slope = 0.11, *p* = 0.15). Consistent with Hypothesis 4a, highly proactive employees engage in voice behavior more when a leader’s proactive personality is low.

With regard to Hypothesis 4b, Figure 2 shows that the interaction of leader and employee proactive personality also significantly predicted taking charge (B = −0.19, SE = 0.06, *p* < 0.001), supporting Hypothesis 4b. The interaction is shown in Figure 4. The slope difference tests showed that the positive relationship between employee proactive personality and taking charge was stronger when leader proactive personality was low (simple slope = 0.35, *p* < 0.001) than high (simple slope = 0.02, *p* = 0.77). Consistent with Hypothesis 4b, highly proactive employees engage in taking charge more when a leader’s proactive personality is low.

Finally, to test Hypothesis 5a,b, we calculated the moderated mediation index. The results shown in Table 3 provide evidence that the leader’s proactive personality moderated not only the indirect effects of employee proactive personality on intrinsic (B = −0.04, SE = 0.02, 95% CI [−0.08, −0.01]) and extrinsic career growth (B = −0.04, SE = 0.02, 95% CI [−0.07, −0.01]) via voice behavior but also the indirect effects of employee proactive personality on intrinsic (B = −0.07, SE = 0.02, 95% CI [−0.11, −0.03]) and extrinsic career growth via taking charge (B = −0.06, SE = 0.02, 95% CI [−0.10, −0.03]). Thus, Hypothesis 5a,b were also supported.

## 5. Discussion

Based on social information processing theory, this study proposed a moderated mediation model to examine the effect of employee proactive personality on intrinsic and extrinsic career growth. The results show that an employee’s proactive personality is positively related to intrinsic and extrinsic career growth. Taking charge plays a mediating role between employee proactive personality and intrinsic and extrinsic career growth by enhancing the positive effect of proactive personality on organizational career growth. Supporting our prediction of the substituting effect of a leader’s proactive personality, we found that a leader’s proactive personality negatively moderates the effects of an employee’s proactive personality on voice behavior and taking charge, as well as the indirect effect of an employee’s proactive personality on intrinsic and extrinsic career growth via taking charge. However, voice behavior is not associated with either intrinsic or extrinsic career growth, nor does it mediate the influence of employee proactive personality on either intrinsic or extrinsic career growth. We suppose that the reason for this finding is that the relationship between voice behavior and organizational career growth forms an inverted U-curve, meaning that up to a certain level, employee proactive personality is beneficial for organizational career growth via voice behavior, but beyond that level, it is detrimental. The quantity and quality of voice behavior beyond this threshold can lead to negative reactions and evaluations from leaders and peers with negative consequences for organizational career growth. This conclusion is supported by the findings of Ng and colleagues (2022), who showed that a certain voice quality leads to workplace ostracism [50]. In addition, employees will exhibit voice behavior and take charge according to the norms and expectations of the organization and, in particular, their leaders. Therefore, a leader’s proactive personality can motivate less proactive employees to adapt and engage in more voice behavior and take charge, thereby realizing organizational career growth.

### 5.1. Theoretical Contributions

Our study contributes to the literature on organizational careers. In response to recent calls for more knowledge about the predictors of organizational career growth [5,6], we focused on the influence of employees’ proactive personality. In contrast to previous studies that mostly treated organizational career growth as a unidimensional variable [2,51], we took a differentiated view of intrinsic and extrinsic career growth, following Weng and Zhu (2020) [1]. Such a multidimensional view of organizational career growth implies that employees improve their professional skills and achieve personal career goals through their own efforts, i.e., realize intrinsic career growth. These efforts are rewarded through promotions and compensation increases, i.e., extrinsic career growth [1,27,47]. While previous research has examined the influence of the Big Five personality traits on organizational career growth [6] or the potential interaction between proactivity and organizational career growth opportunities on individual career outcomes [3,27], our research revealed that employee proactive personality is a significant determinant of organizational career growth. It is an important addition to the literature on organizational career growth, which extends the perspective of theoretical research.

Drawing on social information processing theory, we examined the mediating mechanisms between employees’ proactive personality and intrinsic and extrinsic career growth. Our results show that taking charge plays an important mediating role in this relationship, while voice behavior does not. Although previous studies have mainly examined the conditions of proactive behaviors [16,37,52], few studies have included subsequent career outcomes of these proactive behaviors. Some studies have examined the role of proactive behaviors that are primarily directed toward one’s own career advancement (pro-self proactive behavior), while omitting proactive behaviors that are primarily directed toward the organization and its success (pro-organizational proactive behavior) [9,17,53]. Furthermore, we distinguished two types of proactive behaviors (voice behavior and taking charge) and examined their mediating effects. In this way, our results provide a more comprehensive understanding of the mediating effects of pro-organizational proactive behavior between employee proactive personality and organizational career growth.

Finally, our study also contributes to uncovering a boundary condition of employee proactive personality. Previous studies have mainly focused on the congruence between leader proactive personality and employee proactive personality, rather than using leader proactive personality as a moderating variable [7,54]. In our paper, we demonstrated the substituting effect of leader proactive personality on the relationship between employee proactive personality and organizational career growth. We found that leader proactive personality can substitute the role of employee proactive personality by making less proactive employees engage in proactive behaviors. This effect is consistent with prior research: Hao and Han (2022) confirmed the substitution effect of perceived trust for proactive personality, which also motivates less proactive employees to speak up [16]. Thus, we extend previous studies and the nomological network of proactive personality.

### 5.2. Practical Implications

Our findings also have several practical implications. First, the results of this research are instructive for employees’ own career management and development. Understanding the interplay between proactivity and organizational career growth, employees can create favorable conditions for their career development in organizations by showing initiative. For example, employees can try to take initiative in improving current circumstances or creating new ones, and not passively wait for information and opportunities to come. In this process, employees should learn to improve their personal abilities, accumulate social resources, and achieve self-management of career growth. If employees, in addition, plan their career development for the long term and clearly define their organizational career growth goals, they can increase their chances of succeeding in an increasingly challenging business environment.

Second, the results of this research help organizations better understand the drivers of intrinsic and extrinsic career growth and manage them accordingly to maintain employee satisfaction and prevent talent loss. According to our findings, organizations should not only allow but demand and encourage their employees to engage in various proactive behaviors, which benefit the organization’s functionality and help employees realize their career growth. For example, organizations can attempt to create an open climate and a psychologically safe environment, which encourages employees to take initiative without fear of punishment. And organizations can include proactive behaviors as one of their evaluations toward employees. In addition, since our results show that a leader’s proactive personality can at least partially compensate for an employee’s low proactive personality, organizations and companies can select and develop leaders with a high level of proactive personality, or carry out training to promote leaders’ initiative.

### 5.3. Limitations and Future Research

This study also has limitations. First, although a questionnaire matching method was used, a cross-sectional design still did not allow for strong causal inference. Future studies should use longitudinal or follow-up designs to examine the causal relationship between employee proactive behavior and organizational career growth.

Second, we used a moderated mediation model to test the moderating effect of a leader’s proactive personality. Previous research has pointed out that the way peers perceive employees’ proactivity could affect the initiative’s ability to be effective [55]. So future studies could examine the moderating effect of peer reactions, and other influencing factors, such as the proactive personality of peers. Meanwhile, we did not consider the role of organizational factors (e.g., organizational culture) and task characteristics (e.g., task interdependence). Future research could also explore these important but unexamined factors that may influence employees’ reactions to proactive personality.

Third, this study found that taking charge influences organizational career growth, while voice behavior does not. Regarding the somewhat surprising finding for voice behavior, future research could explore the inverted U-curve explanation. Furthermore, some prior studies have shown that peer support and leadership will affect employee engagement in proactive behavior [24,56]. It would be useful for future research to control these relevant variables to test the incremental validity of employee proactive behaviors in explaining organizational career growth.

Finally, the fact that our sample consisted only of Chinese participants limits the generalizability of the findings. Therefore, we call for future research to investigate whether the findings hold true in cross-cultural comparative studies.

## Figures and Tables

**Figure 1 behavsci-14-00256-f001:**
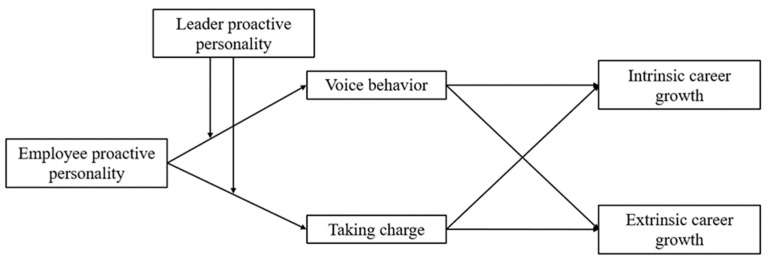
Conceptual model.

**Figure 2 behavsci-14-00256-f002:**
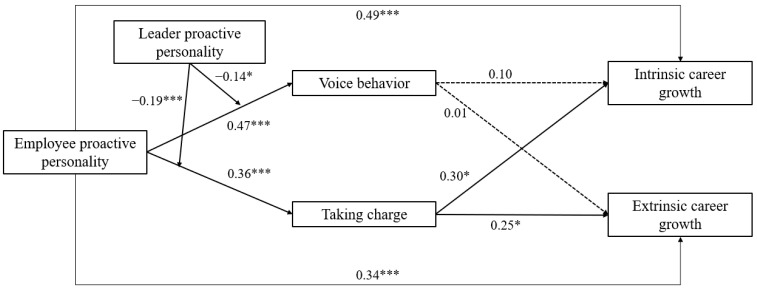
Structural model. Note. * *p* < 0.05, *** *p* < 0.001.

**Figure 3 behavsci-14-00256-f003:**
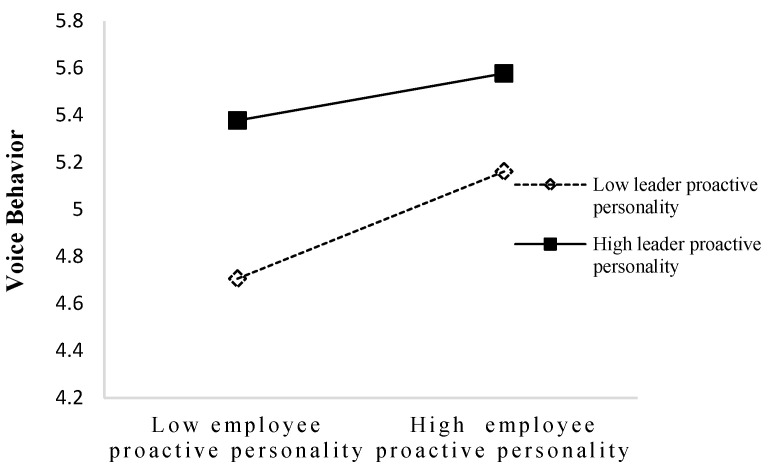
Interaction of employee and leader proactive personality on voice behavior.

**Figure 4 behavsci-14-00256-f004:**
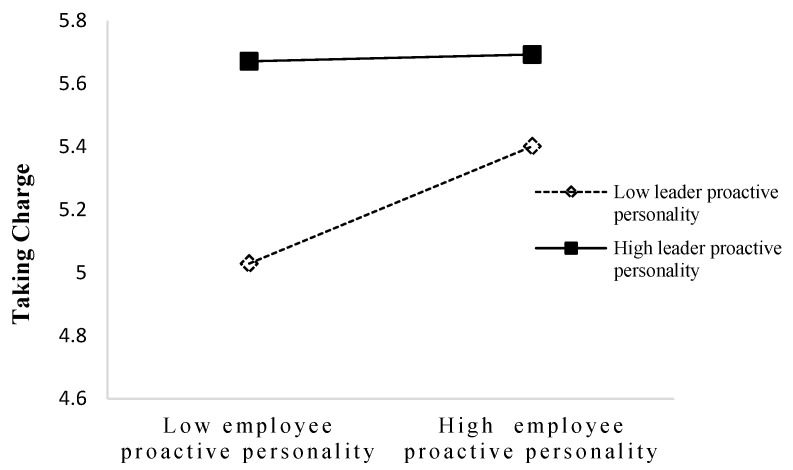
Interaction of employee and leader proactive personality on taking charge.

**Table 1 behavsci-14-00256-t001:** Results from confirmatory factor analyses.

Model	*χ* ^2^	*df*	*χ*^2^/*df*	RMSEA	SRMR	CFI	IFI	TLI
6-Factor model	1582.28	800	1.98	0.06	0.04	0.92	0.93	0.92
5-Factor model ^a^	2126.22	805	2.64	0.07	0.09	0.87	0.86	0.87
4-Factor model ^b^	2581.48	809	3.19	0.09	0.09	0.83	0.82	0.83
3-Factor model ^c^	2859.95	812	3.52	0.09	0.10	0.80	0.79	0.80
2-Factor model ^d^	4350.15	814	5.34	0.12	0.14	0.66	0.64	0.66
One-Factor model	5296.75	815	6.50	0.13	0.14	0.57	0.54	0.57

Note. *N* = 307. RMSEA = Root mean square error of approximation. CFI = Comparative fit index. TLI = Tucker–Lewis index. ^a^ Employee’s proactive personality and leader’s proactive personality combined into a single factor. ^b^ Employee proactive personality, leader proactive personality, and voice behavior combined into a single factor. ^c^ Employee proactive personality, leader proactive personality, voice behavior, and taking charge combined into a single factor. ^d^ Employee proactive personality, leader proactive personality, voice behavior, taking charge, and intrinsic career growth combined into a single factor.

**Table 2 behavsci-14-00256-t002:** Means, standard deviations, and correlations among study variables.

Variables	1	2	3	4	5	6	7	8	9	10	11	12	13	14
1.Region	1													
2.Gender (E)	0.01	1												
3.Age (E)	−0.17 **	0.02	1											
4.Education	−0.03	0.04	0.07	1										
5.Nature	0.10	0.05	−0.13 *	−0.05	1									
6.Tenure	−0.10	0.01	0.55 ***	0.06	−0.12 *	1								
7.Gender (L)	−0.12 *	0.40 **	0.06	−0.01	−0.04	0.08	1							
8.Age (L)	0.05	−0.05	0.18 **	−0.16 **	−0.03	0.08	−0.10	1						
9.EPP	0.08	−0.09	−0.01	−0.01	0.13 *	−0.12 *	−0.07	0.04	(0.88)					
10.LPP	0.06	−0.02	−0.04	−0.13 *	0.14 *	−0.05	−0.13 *	0.14 *	0.31 ***	(0.83)				
11.Voice behavior	0.03	−0.03	0.04	−0.07	0.08	0.11	−0.04	0.05	0.36 ***	0.56 ***	(0.90)			
12.Taking charge	−0.01	0.01	0.05	−0.09	0.06	0.07	−0.02	0.04	0.34 ***	0.54 ***	0.79 ***	(0.95)		
13.Intrinsic career growth	0.07	−0.07	0.01	0.04	0.16 **	−0.08	−0.05	0.08	0.53 ***	0.39 ***	0.46 ***	0.46 ***	(0.93)	
14.Extrinsic career growth	0.05	−0.14 *	−0.04	0.10	0.13 *	−0.11	−0.11 *	0.05	0.45 ***	0.35 ***	0.38 ***	0.39 ***	0.69 ***	(0.89)
Mean	2.62	1.50	2.07	2.79	1.94	3.54	1.39	2.58	4.85	5.23	4.99	5.02	5.10	4.07
SD	2.84	0.50	0.63	0.77	0.86	1.38	0.50	0.68	1.01	0.95	1.14	1.06	1.20	1.17

Note. *N* = 307. (E) and (L) represent employee (E), respectively leader (L) variables. EPP = Employee Proactive Personality. LPP = Leader Proactive Personality. Cronbach’s alpha value of each scale is presented in parentheses. * *p* < 0.05, ** *p* < 0.01, *** *p* < 0.001.

**Table 3 behavsci-14-00256-t003:** Indirect and conditional indirect effects with bootstrapped 95% confidence intervals.

Path	Estimate	SE	95%CI	
			Lower	Upper
indirect effects				
EPP→voice behavior→intrinsic career growth	0.05	0.06	−0.06	0.17
EPP→voice behavior→extrinsic career growth	0.01	0.05	−0.08	0.12
EPP→taking charge→intrinsic career growth	0.11	0.06	0.01	0.23
EPP→taking charge→extrinsic career growth	0.09	0.05	0.01	0.19
conditional indirect effects(LPP)				
EPP→voice behavior→intrinsic career growth	−0.04	0.02	−0.08	−0.01
EPP→voice behavior→extrinsic career growth	−0.04	0.02	−0.07	−0.01
EPP→taking charge→intrinsic career growth	−0.07	0.02	−0.11	−0.03
EPP→taking charge→extrinsic career growth	−0.06	0.02	−0.10	−0.03

## Data Availability

The data presented in this study are available on request from the corresponding author. The data are not publicly available due to confidentiality and research ethics.

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
