# Peer review of "Employee Proactive Personality and Career Growth: The Role of Proactive Behavior and Leader Proactive Personality"

_behavsci, 2024, doi:10.3390/bs14030256_

Round 1

Reviewer 1 Report

Comments and Suggestions for Authors

Thank you for the opportunity to review the manuscript entitled “Employee Proactive Personality and Career Growth: The Role 2 of Proactive Behavior and Leader Proactive Personality”. The study examines whether and how employee proactive personality influences intrinsic and extrinsic career growth, and also the mediating effects of two types of proactive behaviors (voice behavior and taking charge) and the moderating effect of leader proactive personality. Overall, the manuscript is well written, well structured and methodologically rigorous. I have only some minor concerns:

- More information should be reported about the recruitment. Specifically, how companies and participants were contacted? Was an incentive given for participation?

- More information should be reported about measures; were they preliminarily adapted to the Chinese context?

Author Response

Dear Reviewer,

Thank you for your comments on our manuscript entitled “Employee Proactive Personality and Career Growth: The Role of Proactive Behavior and Leader Proactive Personality” (behavsci-2898517). We have carefully considered your comments and made revisions that we hope you will approve. We have highlighted major changes in response to your comments in red in the revised manuscript. Specifically, our responses to your comments are as follows:

  1. Comment: More information should be reported about the recruitment. Specifically, how companies and participants were contacted? Was an incentive given for participation?

Reply: Thank you very much for this comment. According to your suggestion, we added more specifical information about the recruitment in the Participants and Procedures section. The paragraph in the revised version reads as follows: “Before the start of the research, we approached organizational managers in our social networks through telephone, and specifically communicated with them about the study goals, the needed support and cooperation, as well as the distribution method and arrangement of questionnaires. And then, we arrived at companies at the agreed time.” Moreover, to encourage participation and express our appreciation, we provided small gifts to participants as compensation, and we described this part at the end of the second paragraph in the Participants and Procedures section.

  1. Comment: More information should be reported about measures; were they preliminarily adapted to the Chinese context?

Reply: Thank you very much for this comment, which helped us to improve the manuscript. According to your suggestion, we added more information in the Measures section. Furthermore, the scale of organizational career growth was developed by Chinese scholars in Chinese context, and other variables (proactive personality, voice behavior and taking charge) were often used in Chinese context (e.g., Ali et al., 2020; Kim and Liu, 2017; Zhang et al., 2023). Hence, we think these variables were preliminarily adapted to the Chinese context. The revised part has been highlighted in the manuscript.

Ali, A.; Wang, H.; Johnson, R.E. Empirical analysis of shared leadership promotion and team creativity: An adaptive leadership perspective. Journal of Organizational Behavior, 2020, 41(5), 405-423. https://doi.org/10.1002/job.2437

Kim, T.Y.; Liu, Z. Taking charge and employee outcomes: The moderating effect of emotional competence. The International Journal of Human Resource Management, 2017, 28, 775-793. https://doi.org/10.1080/09585192.2015.1109537

Zhang, Y.; Wang, F.; Cui, G.; Qu, J.; Cheng, Y. When and why proactive employees get promoted: A trait activation perspective. Current Psychology, 2023, 42, 31701-31712. https://doi.org/10.1007/s12144-022-04142-3

Once again, thank you very much for your valuable comments, which we feel have helped us in improving the paper.

Reviewer 2 Report

Comments and Suggestions for Authors

The article presents a very important subject for companies and their issues related to career and its relationship with the personality of employees and their superiors. The literature covers the latest developments on the subject, the theoretical framework adopted by the study is appropriate, the analyzes are well conducted and the results well discussed. However, several elements pose serious problems in terms of methodology and data analysis, of which here is one:

1. Adds an analysis of statistical power, especially since the analyzes were conducted using a structural equation model; this requires a larger sample size;

2. The alpha coefficient of variables 10, 11 and 12 is very high; we suspect a problem of redundancy in the items of the measurement scales;

3. In the analyses, it is also appropriate to control the region where the company belongs;

Author Response

Dear Reviewer,

Thank you for your comments on our manuscript entitled “Employee Proactive Personality and Career Growth: The Role of Proactive Behavior and Leader Proactive Personality” (behavsci-2898517). We have carefully considered your comments and made revisions that we hope you will approve. We have highlighted major changes in response to your comments in red in the revised manuscript. Specifically, our responses to your comments are as follows:

  1. Comment: Adds an analysis of statistical power, especially since the analyzes were conducted using a structural equation model; this requires a larger sample size.

Reply: Thank you very much for this comment. According to your suggestion, we added an analysis of statistical power. Specifically, we did a post-hoc power analysis in G*Power. We chose F tests, set alpha(α) to 0.05 and total sample size to 307, and chose a medium effect size. Finally, we calculated a statistical power (1-β) of 0.99, which was generally not less than 0.8. Furthermore, we reviewed relevant literature and found that a sample size of 100 to 150 is the minimum standard for structural equation model. Schumacker and Lomax (2004) pointed out that a sample size of 250 to 500 was used in many studies, and the sample size in this range was appropriate.

Schumacker, Randall E. and Richard G. Lomax (2004). A beginner's guide to structural equation modeling, 2th ed., Mahwah, NJ: Lawrence Erlbaum Associates.

  1. Comment: The alpha coefficient of variables 10, 11 and 12 is very high; we suspect a problem of redundancy in the items of the measurement scales.

Reply: Thank you very much for this comment. The high alpha coefficient may cause common method bias. According to our results in the manuscript, this study did not exist serious common method bias. Meanwhile, both the average variance extracted (AVE) and composite reliabilities (CR) were within reasonable range. These variables have good discriminant validity. Furthermore, according to your suggestion, we reviewed relevant literature and found that previous studies on voice behavior, taking charge and organizational career growth also showed similar results with high alpha coefficient (higher than 0.9; e.g., Jiang et al., 2021; Kim et al., 2023; Lam et al., 2022). So, we think it does not affect the use of variables and the results in this study.

Jiang, Y.; Wang, Q.; Weng, Q.D. Personality and organizational career growth: The moderating roles of innovation climate and innovation climate strength. Journal of Career Development, 2021, 48(4), 521-536. https://doi.org/10.1177/0894845320901798

Kim, S.S.; Pak, J.; Son, S.Y. Do calling-oriented employees take charge in organizations? The role of supervisor close monitoring, intrinsic motivation, and organizational commitment. Journal of Vocational Behavior, 2023, 140, 103812. https://doi.org/10.1016/j.jvb.2022.103812

Lam, C.K.; Johnson, H.H.; Song, L.J.; Wu, W.; Lee, C.; Chen, Z. More depleted, speak up more? A daily examination of the benefit and cost of depletion for voice behavior and voice endorsement. Journal of Organizational Behavior, 2022, 43(6), 983-1000. https://doi.org/10.1002/job.2620

  1. Comment: In the analyses, it is also appropriate to control the region where the company belongs.

Reply: Thank you very much for this comment. We agree with your view and added the region where the company belongs as a control variable. Meanwhile, we did the calculation again, and found that the results had not changed much and the test results of hypotheses remained the same. The revised part has been highlighted in the manuscript.

Once again, thank you very much for your valuable comments, which we feel have helped us in improving the paper.

Reviewer 3 Report

Comments and Suggestions for Authors

Dear authors

Thank you for the opportunity to review this paper that I found interesting, especially on a theoretical level.

I have a few comments that I think could improve the paper

1. In the introduction, you rely a lot on the work of Weng. It would be nice to have other sources too, to expand the scope of the theoretical foundation of the main concepts you use'

2. Description of the participants could be elaborated, especially in a better description of what kind of companies were involved in terms of size, business, etc.

3. Lines 281-287 should be moved to the results section

4. In the discussion section, I miss a discussion of the interrelatedness of the main constructs used in the study. Is it possible that employee behavior and personality is linked to, e.g., the type of company, overall work environment (leadership, coworkers, tasks, etc.), affecting the results of the studyThe discussion section would in general benefit from a more wider angle on the applicability of the results, especially because this study is cross-sectional.   

Author Response

Dear Reviewer,

Thank you for your comments on our manuscript entitled “Employee Proactive Personality and Career Growth: The Role of Proactive Behavior and Leader Proactive Personality” (behavsci-2898517). We have carefully considered your comments and made revisions that we hope you will approve. We have highlighted major changes in response to your comments in red in the revised manuscript. Specifically, our responses to your comments are as follows:

  1. Comment: In the introduction, you rely a lot on the work of Weng. It would be nice to have other sources too, to expand the scope of the theoretical foundation of the main concepts you use.

Reply: Thank you very much for this comment. According to your suggestion, we reviewed relevant literature and found other sources. For instance, Vande Griek et al (2020) and Spagnoli (2020) have also constructed research frameworks around organizational career growth and explored the importance of organizational career growth. Hence, we added other sources (e.g., Spagnoli, 2020; Vande Griek et al., 2020) in the Introduction section.

Spagnoli, P. Organizational socialization learning, organizational career growth, and work outcomes: A moderated mediation model. Journal of Career Development, 2020, 47(3), 249-265. https://doi.org/10.1177/0894845317700728

Vande Griek, O.H.; Clauson, M.G.; Eby, L.T. Organizational career growth and proactivity: A typology for individual career development. Journal of Career Development, 2020, 47(3), 344-357. https://doi.org/10.1177/0894845318771216

  1. Comment: Description of the participants could be elaborated, especially in a better description of what kind of companies were involved in terms of size, business, etc.

Reply: Thank you very much for your comment. According to your suggestion, we added more specifical information about the description of the participants in the Participants and Procedures section. The paragraph in the revised version reads as follows: “A sample was drawn from different companies mainly in five regions of China (Shaanxi, Shanghai, Zhejiang, Guangdong, and Beijing). To ensure the universality of the sample, we selected companies from all walks of life, including state-owned enterprises, private enterprises, wholly foreign-owned enterprises and so on.”

  1. Comment: Lines 281-287 should be moved to the results section.

Response: Thank you for your comment. According to your suggestion, we referred to the relevant literature in this journal (e.g., Wang et al., 2023), and moved Lines 281-287 to the results section. The revised part has been highlighted in the manuscript.

Wang, H.; Wu, S.; Wang, W.; Xiao, Y. Left-Behind Experiences and Cyberbullying Behavior in Chinese College Students: The Mediation of Sense of Security and the Moderation of Gender. Behav. Sci. 2023, 13, 1001. https://doi.org/10.3390/bs13121001

  1. Comment: In the discussion section, I miss a discussion of the interrelatedness of the main constructs used in the study. Is it possible that employee behavior and personality is linked to, e.g., the type of company, overall work environment (leadership, coworkers, tasks, etc.), affecting the results of the study? The discussion section would in general benefit from a more wider angle on the applicability of the results, especially because this study is cross-sectional.

Response: Thank you very much for this comment. We are sorry that we have only a limited discussion of colleague reactions as boundary conditions and the exploration of the inverted U-curve between voice behavior and organizational career growth. According to your suggestion, we combined with previous relevant literature and added related discussion about the overall work environment which may affect the results of the study in the Limitations and Future Research section. The revised part has been highlighted in the manuscript.

Once again, thank you very much for your valuable comments, which we feel have helped us in improving the paper.